# Lytic Bacteriophage Is a Promising Adjunct to Common Antibiotics across Cystic Fibrosis Clinical Strains and Culture Models of *Pseudomonas aeruginosa* Infection

**DOI:** 10.3390/antibiotics12030593

**Published:** 2023-03-16

**Authors:** Isaac Martin, Sandra Morales, Eric W. F. W. Alton, Jane C. Davies

**Affiliations:** 1National Heart and Lung Institute, Imperial College London, Emmanuel Kaye Building, London SW3 6LY, UK; 2Royal Brompton Hospital, Part of Guy’s and St. Thomas’ Trust, Sydney St., London SW3 6NP, UK; 3Department of Paediatrics and Translational Medicine, The Hospital for Sick Children, University of Toronto, Toronto, ON M5G 1X8, Canada; 4AmpliPhi Biosciences, Brookvale, NSW 2100, Australia

**Keywords:** bacteriophage, *Pseudomonas aeruginosa*, antimicrobial resistance, cystic fibrosis, novel antimicrobials, adjunctive therapy, pulmonary infection

## Abstract

Bacteriophages (phages) are antimicrobials with resurgent interest that are being investigated for the treatment of antibiotic refractory infection, including for *Pseudomonas aeruginosa* (Pa) lung infection in cystic fibrosis (CF). In vitro work supports the use of this therapy in planktonic and biofilm culture models; however, consistent data are lacking for efficacy across different clinical Pa strains, culture models, and in combination with antibiotics in clinical use. We first examined the efficacy of a 4-phage cocktail as an adjunct to our CF centre’s first-line systemic combination antibiotic therapy (ceftazidime + tobramycin) for 16 different clinical Pa strains and then determined subinhibitory interactions for a subset of these strains with each antibiotic in planktonic and biofilm culture. When a 4-phage cocktail (4 × 10^8^ PFU/mL) was added to a ceftazidime-tobramycin combination (ceftazidime 16 mg/mL + tobramycin 8 mg/mL), we observed a 1.7-fold and 1.3-fold reduction in biofilm biomass and cell viability, respectively. The four most antibiotic resistant strains in biofilm were very susceptible to phage treatment. When subinhibitory concentrations of antibiotics and phages were investigated, we observed additivity/synergy as well as antagonism/inhibition of effect that varied across the clinical strains and culture model. In general, more additivity was seen with the phage-ceftazidime combination than with phage-tobramycin, particularly in biofilm culture, where no instances of additivity were seen when phages were combined with tobramycin. The fact that different bacterial strains were susceptible to phage treatment when compared to standard antibiotics is promising and these results may be relevant to ongoing clinical trials exploring the use of phages, in particular in the selection of subjects for clinical trials.

## 1. Introduction

In cystic fibrosis (CF) lung infection, antibiotic administration has contributed to increased longevity and improved quality of life [1]. In cases of chronic infection, especially with *Pseudomonas aeruginosa* (Pa), this requires chronic daily suppressive therapy nebulised into the lungs [2,3]. Unfortunately, this daily nebulisation rarely clears well-established infection and the disease course of patients with CF and chronic infection is punctuated by periods of exacerbation that require further courses of oral, inhaled or intravenous antibiotics.

Within the CF lung, chronic Pa exists as aggregates that are suspended within the sputum matrix [4,5]. This biofilm lifestyle contributes to the survival of the bacteria by providing physical protection of the bacteria from immune cells and antimicrobial therapies. Additionally, the heterogeneity of the structure results in reservoirs of cells with differential access to nutrients, resulting in differing metabolic profiles and rates of cell division and this also impacts antibiotic treatment. It is known that the minimum inhibitory concentration (MIC) of planktonically grown bacteria can be many orders of magnitude higher for the same bacteria when grown in biofilms [6]; however, it is often impossible to achieve these concentrations in the airway [7,8,9].

There is an expanding body of evidence demonstrating the potential of bacteriophage (phage or phages) to combat Pa infection in CF lungs. We have previously shown increased survival and lowered inflammatory response in a model of murine Pa lung infection [10] and demonstrated phage sensitivity over time for many MDR strains (including the Liverpool Epidemic Strain) in patients at our hospital [11]. Although these and other studies provide a promising evidence base, as this therapy makes the leap from laboratory to human therapy in several ongoing human trials [12,13,14,15], there is no reason that the interactions between antibiotics, phage and bacteria should be consistent across the concentrations or type of bacterial growth encountered in the lung.

There is a body of preclinical experimentation investigating the effects of phage-antibiotic combinations in different culture systems. One study examining 17 clinical Pa strains with documented resistance to phage in a planktonic assay, demonstrated phage-replication within the biofilms of eight strains (47%) of the planktonically-resistant strains [16]. One study found that phage virion production was blocked with the administration of twice the minimum inhibitory concentration (MIC) of rifampicin in a *Ralstonia solanacearum* culture [17]. There are also Pa specific examples of phage-antibiotic antagonism that highlight this as a risk. One early study revealed that burst sizes (the amount of phage virion progeny released at cell lysis) of anti-Pseudomonal phage PM2 was reduced to less than 1% of that seen in control conditions with simultaneous addition of rifampicin at a concentration of 0.3 μg/mL [18]. Subsequent work with Pa decades later demonstrated that subinhibitory concentrations of rifampicin completely inhibited phage (lytic phages PAK_P3 and PAK_P4) replication [19].

There is, however, a similar body of in vitro work supporting phage additivity or synergy. Gu Liu et al. (2020) recently looked at many different antibiotic classes combined with phages against a highly drug-resistant *E. coli* in a checkerboard assay and found that phages could render this MDR bacteria susceptible to antibiotics at concentrations to which it had previously demonstrated resistance [20]. These experiments demonstrated synergy with phage-ceftazidime combination and additivity of effects with phage-kanamycin, an aminoglycoside antibiotic in the same class as tobramycin.

Tobramycin (TOB) is an aminoglycoside antibiotic that targets the bacterial ribosome 30S subunit, and ribosomal function is essential for virion replication and the cell lysis of bacteriophage [21]. Ceftazidime (CTZ), a ß-lactam, works at the cell wall, which is where bacteriophages adsorb to surface proteins, necessary for allowing entry of genetic material into the cell [21]. Exposing Pa to an antibiotic to which it is susceptible can, in theory, block subsequent phage infection and phage-mediated killing. Many classes of antibiotics interfere with bacterial cell physiology in ways that are crucial for phage infection, and the possibility of non-additive or even inhibitory/antagonistic results must be considered.

Here, we wished to explore the efficacy of a 4-phage cocktail that has been used in several preclinical studies and for compassionate use [22,23,24,25] against Pa isolates from CF airway samples. We used two common anti-Pseudomonal antibiotics: ceftazidime and tobramycin. In the first experiments, we use a high-concentration combination of these two agents and assess the extra biofilm reduction with the addition of phage in 16 different Pa clinical strains. In the subsequent experiments, we investigate the effects of subinhibitory concentrations of phage cocktail along with subinhibitory concentrations of ceftazidime and tobramycin individually, in both biofilm and planktonic culture for 10 different strains.

## 2. Results

### 2.1. Assessing Biofilm Eradication with Phage with an Antibiotic Mix across 16 Clinical Strains

Results of the resazurin assay (cell viability) and the crystal violet (CV) assay (biomass) correlated well with each other (Pearson coefficient 0.80, *p* = 0.0002). The level of planktonic sensitivity to bacteriophage treatment did not correlate with the outcomes of either biofilm assay (Pearson’s coefficient of 0.36 and 0.41 for the resazurin and CV assays, respectively). We concluded that the degree of planktonic sensitivity to phage activity did not predict biofilm eradication. These results are summarized in Figure 1.

In general, the results of the plaque assay were not predictive of the amount of biofilm reduction. The results depended on the clinical strain and also on the type of assay being used, whether assessing cell viability or biomass. Across the group of 16 strains, bacteriophage cocktail monotherapy at an apparent titre of 4 × 10^7^ PFU/mL resulted in a decrease in cell viability of 46% (95% CI 28.0–64.6%) and biomass of 59% (95% CI 43.0–74.7%) and these means they were not significantly different from one another.

The addition of phages enhanced the anti-biofilm efficacy of a fixed ceftazidime (CTZ)—tobramycin (TOB) combination across clinical strains (Figure 2).

The four strains that demonstrated the greatest level of biofilm reduction with the addition of the phage cocktail (strains 1, 5, 7 and 14) were the most antibiotic resistant in biofilm culture when assessed by the CV assay (all <10% reduction in biofilm biomass).

### 2.2. Assessing Subinhibitory Concentrations of Phage and Antibiotics in Planktonic and Biofilm Culture

Ten (10) randomly selected strains from the initial 16, as well as laboratory strain PA01 underwent further analysis in planktonic and biofilm culture at subinhibitory phage and antibiotic combinations. Strain 16 was excluded on the basis that it was sometimes resistant to phage in plaque assay, but not at time point 1, which was when it was included in the first set of experiments. A summary of these findings is represented in Table 1.

With planktonic cultures, combination antibiotic-phage therapy resulted in a higher rate of bacterial eradication 54.5% of the time (95% CI; 34.7–73.1%), whereas in biofilm culture this was only seen 27.3% of the time (95% CI; 13.2–48.2%). For individual antibiotics, combination ceftazidime-phage therapy resulted in a statistically higher rate of bacterial eradication 59.1% of the time (95% CI; 38.7–76.7%), whereas for tobramycin this was 22.7% (95% CI; 10.1–43.4%). Notably, in biofilm, there were no strains that demonstrated significant additivity of effect with phage-tobramycin therapy, whereas 6/11 strains tested (54.5%) demonstrated additivity or synergy with ceftazidime-phage combination therapy. Strain PA01 with ceftazidime-phage in biofilm culture, coefficient of drug interaction (CDI) = 0.44 ± 0.10; Strain 7 with tobramycin-phage in planktonic culture, CDI = 0.43 ± 0.07; and Strain 11 with ceftazidime-phage in biofilm culture, CDI = 0.58 ± 0.12. Two of the 3 instances in which phage-antibiotic antagonism was seen were due to reduced efficacy of phage when combined with the antibiotic. There was one instance where combination treatment led to reduced antibiotic efficacy (Strain 5, tobramycin-phage in biofilm depicted in Figure 3d).

We also investigated whether planktonic and biofilm responses to the addition of phage correlated with one another. For the 4 conditions in Table 1 (N, A, I, S), we assigned a value of 0 for N, −1 for I, 1 for A and +2 for S. Results in either assay did not correlate with one another (Spearman correlation, *p* = 0.22).

Representative examples of all possible outcomes in Table 1. (Synergy, additivity, inhibition and no added effect) are shown in Figure 3.

## 3. Discussion

We have demonstrated that the addition of a lytic phage cocktail to a combination of ceftazidime and tobramycin, two anti-Pseudomonal antibiotics commonly used together in clinical practice, significantly reduced biomasses and viability of biofilm-grown clinical Pa strains from CF airway cultures. Encouragingly, the four strains demonstrating the highest level of resistance to the antibiotic combination therapy demonstrated high levels of sensitivity to the bacteriophage cocktail.

We then looked at the efficacy of subinhibitory concentrations of phage when combined with subinhibitory concentrations of ceftazidime or tobramycin in a checkerboard assay and found that there was more additivity/synergy when phage was combined with ceftazidime, as opposed to tobramycin, especially in biofilm culture. It is, worth noting that there were no instances of additivity seen in biofilm culture with tobramycin for the 10 clinical strains examined. There were instances of both synergy and inhibition for ceftazidime in both planktonic and biofilm cultures. This research is important in that it shows efficacy in an array of clinical strains found in a specific disease process alongside clinically relevant antibiotics and concentrations across different culture models (planktonic and biofilm).

Ultimately, the final message from these experiments is one of variability. Individual clinical strains interact differently with antibiotics and phages in different growth conditions and we were not able to predict responses to these combinations. To this point, we found early on that planktonic sensitivity to phage did not predict efficacy of biofilm reduction. There were instances of poor planktonic response to phages but a strong reduction in biofilm, as well as exquisite sensitivity to the phage in planktonic culture but a poor response in biofilm. Furthermore, our checkerboard experiments for individual antibiotic-phage combination therapy showed poor correlation between planktonic and biofilm culture models.

One of the explanations for additivity is that the 2 modalities target different populations, both for reasons of differential access to cells and due to the mechanism of action. For instance, in contrast to most conventional antibiotics which require actively dividing bacterial cells, bacteriophage can lyse metabolically dormant cells. A study investigating *S. aureus* phages demonstrated infection and lysis of metabolically dormant persister cells in addition to exopolysaccharide degradation of the biofilm matrix [26]. This finding is promising for an adjunctive therapy as it indicates that the agents would not necessarily be competing for access and killing the same cell populations, but might be broadening the spectrum of populations eradicated. With regards to cells in different geographic locations within the biofilm, it is, known that some lytic phages specific to Pa employ the use of exopolysaccharide degrading enzymes that can disperse biofilms, which may prove beneficial in chronic, biofilm infections [27]. The use of such phage-encoded exopolysaccharide depolymerases has been put forth as promising antibiotic adjuvants in their own right [28]. Thus, one hope is that by using agents that work in different ways, the success of current antimicrobial therapy would be increased and that this may translate to improved patient outcomes in CF.

There has been a large focus in the literature on conditions that inhibit the action of phages, and such studies have largely looked at the application of antibiotics prior (i.e., as pre-treatment) to the administration of phage [17,19]. There are indeed studies that suggest the order of application can significantly alter the outcome of antibiotic-phage experiments. Several studies involving both *Staphylococcus aureus* and Pa have shown that when phage treatment was administered prior to antibiotics, results were additive or synergistic, as opposed to when phage was administered after antibiotics when results were seen to be antagonistic to the action of phage [29,30]. Although the mechanism of this phage antagonism is unclear, some hypotheses that have been put forth are a reduction in bacterial density leading to a reduction in viral replication, as well as the possibility that antibiotics directly interfere with phage infection and/or replication within the cell. Notably, Chaudhry et al. (2017) found that a delay of 24 h between aminoglycoside administration (gentamicin or tobramycin) and phage in an in vitro Pa model of infection effectively suppressed any phage replication [30].

When designing our experimental protocol, the timing of phage and antibiotic administration had been intended to be simultaneous; however, in practice, antibiotic administration was just before the addition of phage and time lags between antibiotic and phage administration could have been as high as 10 min for individual plates. In this way, we may have been unintentionally reducing the efficacy of the phage in both planktonic and biofilm culture, giving the antibiotics time to exert effects leading to antagonism of phage. This type of antagonism (antibiotics lessening the effects of phage) was seen in 2 experimental conditions: once in planktonic culture with ceftazidime and once in biofilm culture with tobramycin; Table 1). Notably, in only 1 out of the 3 instances of antagonism seen did the combined administration of phage worsen the efficacy of antibiotics alone (this was seen in biofilm culture with tobramycin in Strain 5, Figure 3d). This example is noteworthy not only because it was the only case of *antibiotic* antagonism that was seen across the 10 strains, but because the results in planktonic culture were seen to be additive with tobramycin. The reasons for this antagonism are not clear, but notably, the assay used (CV) does not give an approximation of cell viability and it could have simply been more dead matter that had adhered to the well. Although the results of the CV and resazurin assays correlated well, this was not true for every strain.

An obvious criticism of any in vitro work is that it does not adequately simulate the complexity of bacterial infection in a living organism. The fact that bacteria exist in polymicrobial communities within the lung environment is one such factor [31]. This is thought to explain some of the disconnects between laboratory-based methods of antibiotic sensitivity testing (both in planktonic and in biofilm cultures) compared to clinical outcomes [32]. In one experiment using *E. coli* and Pa, the corresponding phage could eradicate each monoculture biofilm, whereas the two bacteriophages applied together were not effective in a mixed biofilm [33]. To address this, further in vitro work needs to be carried out in different culture models and conditions that more accurately simulate the lung environment, such as mixed species biofilms, artificial sputum media and/or small animal models of CF.

## 4. Materials and Methods

### 4.1. Selection and Storage of Pa Strains

One laboratory strain (PA01) was used in these experiments [34]. For clinical isolates, a Microsoft Access database of Pa from CF patients at the Royal Brompton Hospital (RBH) was used. Clinical Pa strains were provided by the RBH microbiology laboratory and had been identified by matrix-assisted laser desorption/ionization-time of flight (MALDI-TOF) analysis as being Pa [35]. Samples from CF patients were taken directly from sweeps of bacterial colonies grown on agar with a plastic inoculation loop. Strains were stored on glass Microbank™ beads (ProLab Diagnostics, Richmond Hill, ON, Canada) in a −80 °C freezer. Colonies were seeded on nutrient agar for overnight growth in the incubator at 37 °C.

Isolates were selected based on recorded sensitivity to both ceftazidime and tobramycin by disc diffusion assay [36].

### 4.2. Plaque Assay

The phages used in this study were provided by AmpliPhi Australia Pty Ltd. (Brookvale, NSW 2100, Australia). AB-PA01 is a combination of four lytic bacteriophages, Pa193 and Pa204 belonging to the Myoviridae family, and Pa222 and Pa223 belonging to the Podoviridae family. The titer for the phage mix on the company’s *P. aeruginosa* strains used for these experiments was 4 × 10^8^ PFU/mL. The phage titre was determined using plaque assay, as described below. None of these phage components encoded any known bacterial virulence or antibiotic resistance genes, and all phages were considered to be strictly lytic as per the company’s published procedures [25]. The phages were produced following the current good manufacturing practice standard (cGMP) and approved by the US Food and Drug Administration as an investigational new drug for an expanded access program (NCT03395743) [37]. An updated release of AB-PA01 with a higher titre has been used in preclinical studies and compassionate use cases [24,38].

We performed the standard plaque assay for bacteriophage sensitivity testing, as previously described [39]. In brief, an overnight culture of a selected Pa isolate was placed in nutrient broth and grown overnight. Overnight bacterial culture was then diluted in fresh nutrient broth to an optical density (OD) of 0.05. One hundred μL was added to 3 mL of semi-solid nutrient agar (Oxoid, Basingstoke, UK), poured over sterile nutrient agar plates (Oxoid, UK) and allowed to dry in the safety cabinet for 20–30 min. Five (5) μL spots of each phage dilution (neat phage to 10^−6^ dilution) were spotted onto the surface of the Pa inoculated plates and then incubated overnight (16–24 h) at 37 °C before assessing for plaques on the bacterial lawn.

All isolates were examined by plaque assay and screened for their sensitivity to at least the neat bacteriophage cocktail (4 × 10^8^ PFU/mL). Sixteen (16) different Pa isolates were chosen for the first section of experiments and 10 of these were selected for use in the antibiotic-phage checkerboard experiments.

### 4.3. Biofilm Assays

To assess biofilm reduction, we used 96-well plates for two different, high-throughput assays: crystal violet (CV) to assess biofilm biomass [40] and resazurin salt (Alamar Blue, IUPAC name: 7-hydroxy-10-oxiphenoxazin-10-ium-3-one) to assess cell viability [41].

Two hundred (200) μL of overnight culture shaken at 200 rpm at 37 °C in nutrient broth (Oxoid) was diluted to an OD of 0.05 and then pipetted into each well of a 96 well plate which was incubated at 37 °C for 24 h static. The plates were sealed with a membrane to reduce evaporation, but allow gas exchange (Axygen, Sigma Aldrich, Gillingham, UK). Wells were then emptied by pipetting and washed with a phosphate-buffered saline (PBS) solution before 180 µL of Mueller Hinton broth (MHB) (Sigma Aldrich) was added to all wells. Twenty (20) μL of each treatment was applied to the wells. There were three different treatment protocols:

(i) An antibiotic mixture of ceftazidime and tobramycin, both at 2× the published breakpoints for sensitivity (final concentration in wells was 16 mg/mL ceftazidime + 8 mg/mL of tobramycin);

(ii) The bacteriophage cocktail on its own (final concentration 4 × 10^7^ PFU/mL or 1 × 10^6^ PFU/phage); and

(iii) The antibiotic combination of ceftazidime and tobramycin as described above + the phage cocktail (final concentration 4 × 10^7^ PFU/mL of phage, 16 mg/mL ceftazidime and 8 mg/mL tobramycin).

Plates were then incubated overnight (a further 16–20 h) at 37 °C before final biofilm measurements were obtained.

To assess cell viability (resazurin), all wells were emptied of free-floating cells by microchannel pipette and washed with PBS before the introduction of 180 μL of MHB, then 20 μL of resazurin 0.02% *w*/*v* working solution. Plates were covered with aluminium foil and incubated at 37 °C for 3–4 h before being transferred to the plate reader for analysis using emission and excitation wavelengths of 540 and 590 nm, respectively (FI540/590).

To assess biofilm biomass (CV), plates were emptied by tipping out contents and washed twice by submersion and inversion in water before the remaining liquid was removed by inversion. One hundred eighty (180) μL of 0.1% crystal violet solution was added to each well for 15 min before 2 further inversion and washing steps in water. Two hundred (200) μL of 95% ethanol was then added to each well to solubilise the biofilms and then left for 30 min before absorption readings with the spectrophotometer (OD550).

### 4.4. Checkerboard Assay

The checkerboard assay has been previously described in detail, whereby serial dilutions of two different antimicrobial compounds are administered to bacterial culture in a 96-well plate [42,43].

Concentrations for ceftazidime and tobramycin ranged from 1/16th to 8× the published EUCAST minimum inhibitory concentration (MIC) values [36]. For phage, concentrations ranged from 4 × 10^2^ to 4 × 10^8^ PFU/mL. To optimally appreciate the effects of combination therapy in planktonic and biofilm culture, for antibiotics and phages, we used the first dilution below which resulted in either >75% reduction in planktonic growth as assessed by OD or >75% reduction in biofilm biomass as assessed by CV assay. For antibiotics, this work was done in 2-fold serial dilutions and for bacteriophage, we worked with 10-fold serial dilutions. For biofilm culture, there were instances where the highest antibiotic and phage concentrations did not achieve >75% biofilm eradication. In these instances, we assessed the highest concentration tested.

For planktonic culture, 180 μL of overnight culture was diluted in Mueller Hinton broth (Sigma Aldrich) to an OD of 0.1 and then pipetted into each well of a 96-well plate. Twenty (20) μL of solution for treatment groups was administered so that final target concentrations were reached and final volume in the wells = 200 μL. For negative control wells, PBS was used as this was used for phage and antibiotic dilutions.

Results were evaluated spectrophotometrically at 18 h after treatment overnight in the incubator at 37 °C shaking at 200 rpm. Treatment groups (n = 6 technical replicates each) were compared to each other and standardized across strains to account for differences in OD measurements according to the following equation:Percent reduction (%) in OD = ((OD_control/no treatment_ − OD_treatment_)/OD_control/no treatment_) × 100

For biofilm assays, 200 μL of overnight culture was diluted to an OD of 0.1 in Nutrient broth and then pipetted into each well of a 96-well plate. Plates were incubated at 37 °C static for 24 h to allow cells to adhere to the sides and bottom of the wells. Wells were then emptied and washed with a phosphate-buffered saline (PBS) solution before 180 μL of Mueller Hinton broth (Sigma Aldrich) was added to all wells along with different treatments so that the final volume in the wells = 200 μL. For negative control wells, PBS was used as this had been used for phage and antibiotic dilutions. Plates were then incubated at 37 °C overnight (a further 16–20 h) shaking at 200 rpm before final biofilm measurements were obtained using the crystal violet assay, as described earlier in the methods section. Treatment groups consisting of 6 technical replicates were compared to each other and standardized across strains to account for differences in biomass measurements according to the following equation:Reduction (%) of biomass with CV assay = ((OD_control/no treatment_ − OD_treatment_)/OD_control/no treatment_) × 100

### 4.5. Coefficient of Drug Interaction

A coefficient of drug interaction (CDI) was developed for the assessment of the interaction between two different antimicrobial compounds and was used per previously published data [44]. It is calculated based on the following formula:CDI = AB/(A × B)

According to the OD readings for each treatment group, AB is the ratio of the antibiotic-phage combination groups to the negative control group; A or B is the ratio of the single agent (antibiotic or phage) group to the negative control group. A CDI value of <1 indicates that the combination of A + B is synergistic. For these experiments, we have been conservative and used CDI values of <0.7 with standard deviations that do not cross 1 as synergistic.

### 4.6. Statistical Analysis

All statistical analysis and graphs were performed using GraphPad Prism software (Boston, MA, USA). Tests of normality were performed for all data using the Shapiro–Wilk test unless otherwise stated and data were said to be normally distributed if an alpha of 0.05 was reached. For all work in the microtiter plates, ‘robust regression followed by outlier elimination’ (ROUT) was employed to detect outliers if the data were normally distributed (false discovery rate below 1%) [45].

For the first experiments, data across all 16 clinical strains were non-parametric and the Wilkoxon signed-rank test was used for comparison between treatment groups. For the checkerboard assay, one-way ANOVAs were performed to compare outcomes between treatment groups, correcting for multiple comparisons. If the mean effects of combination treatment groups for the 3 biological replicates were statistically different using one-way ANOVAs, CDIs were calculated to determine whether the combination was synergistic (S) or simply additive (A). Experimental conditions in which the effect of combination therapy was significantly less than either single phage or antibiotic therapy were classified as inhibitory (I). Where the results of group means were not statistically different, the combination treatment was said to have no effect (N).

For comparing the relative success of combination therapy based on antibiotic choice/type of culture, classification was binomial: success (additive or synergistic) or failure (no effect or antagonistic). The modified Wald method was used to calculate confidence intervals of the proportion of successful outcomes for ceftazidime and tobramycin in combination with phage in both planktonic and biofilm cultures [46].

## 5. Conclusions

The 4-phage cocktail used in these experiments showed efficacy across a variety of clinical Pa strains both planktonically and within biofilm models. Notably, it was first shown to improve the eradication of pre-formed biofilms when combined with ceftazidime and tobramycin. In further experiments, subinhibitory phage-antibiotic combinations were seen to be synergistic/additive, having no statistically discernable effect, or being antagonistic/inhibitory.

Although it is difficult to draw conclusions that might affect the interpretation of clinical trials exploring phage therapy that are currently in progress, it is important that further in vitro work be carried out to investigate these interactions, similar to work that has been carried out exploring antibiotic-phage combinations in other organisms [20,47]. This is particularly valid based on our finding that the aminoglycoside tobramycin was less likely to result in a favourable outcome when combined with phages. For the clinical trials that are currently being conducted, it would be reasonable to investigate the timing of antibiotic administration in relation to phage administration or perform sub-group analyses in patients to determine if there are any clinical effects observed for patients on concurrent antimicrobial therapy—especially those on chronic suppressive therapy with aminoglycosides.

## Figures and Tables

**Figure 1 antibiotics-12-00593-f001:**
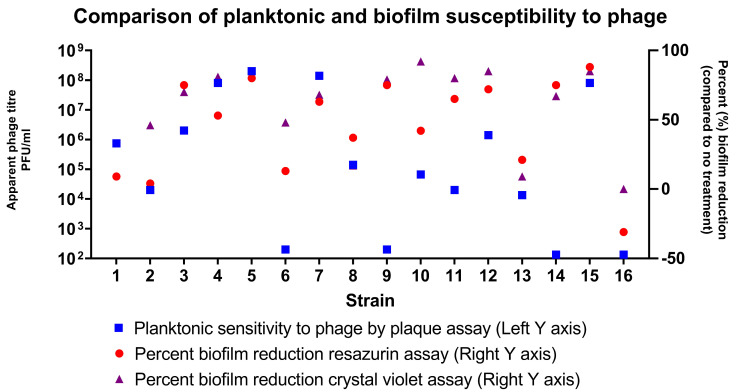
Comparison of planktonic and biofilm susceptibility to phages. Planktonic sensitivity to phages as assessed by plaque assay (blue square) is shown on the logarithmic scale on the left-sided *Y* axis. The per cent reduction in biofilm cell viability (red circle—assessed by immunofluorescent resazurin assay) and biofilm biomass (purple triangle—assessed by crystal violet OD readings) are plotted on the right-sided *Y* axis.

**Figure 2 antibiotics-12-00593-f002:**
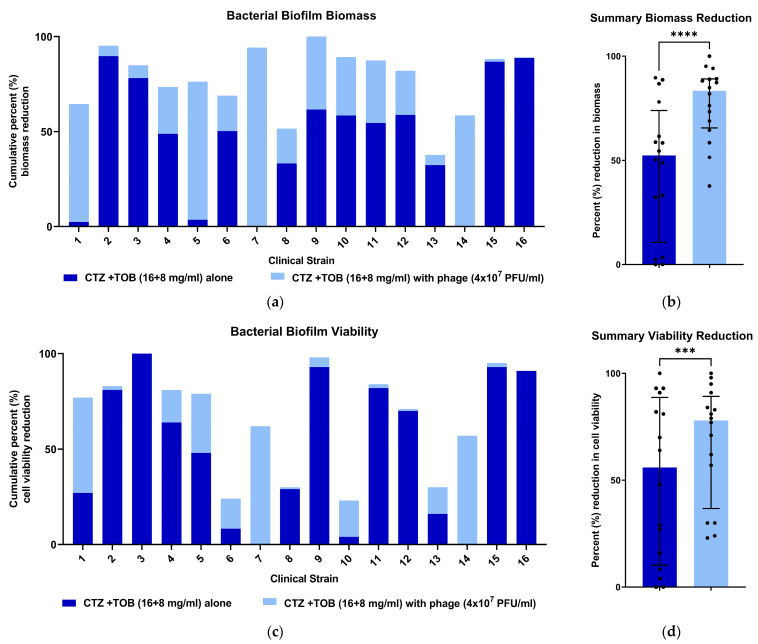
The percent reduction in biofilm biomass and cell viability with antibiotic combination and phage. Dark blue shows the amount of biofilm reduction with the antibiotic combination alone, with the added light blue showing the additional reduction in killing when the bacteriophage was added. The bars, when shown, represent the median and interquartile range. (**a**) The percent reduction of biofilm biomass when the phage cocktail was added to a CTZ + TOB antibiotic mixture for each strain. (**b**) The addition of the phage cocktail resulted in a further 1.7-fold reduction in biofilm biomass across the 16 strains (****, *p* < 0.0001 using the Wilcoxon signed rank test) compared with the dual antibiotics alone. (**c**) The percent reduction in metabolic activity of biofilm cells when the phage cocktail was added to a CTZ + TOB antibiotic mixture. (**d**) The addition of the phage cocktail resulted in a further 1.3-fold reduction in cell viability across the 16 strains (***, *p* < 0.001 using the Wilcoxon signed rank test) compared with the dual antibiotics alone.

**Figure 3 antibiotics-12-00593-f003:**
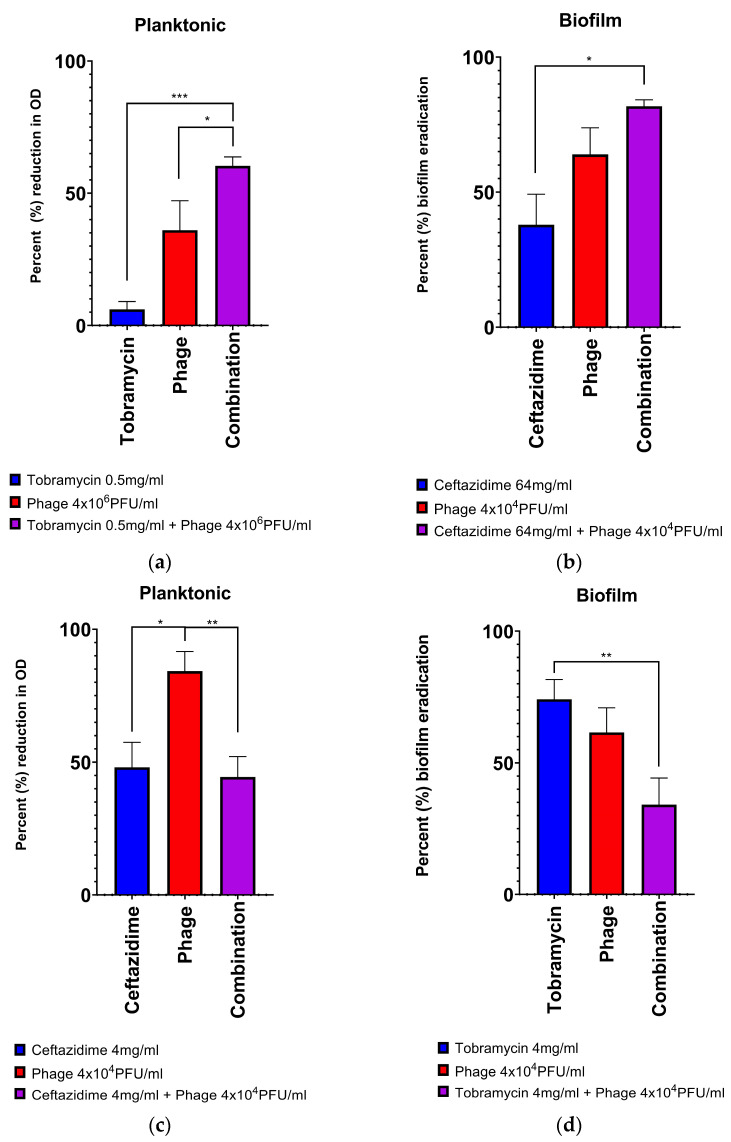
Examples of all different classes of phage-antibiotic combinations: (**a**) synergy, (**b**) additivity, (**c**) inhibition of phage by antibiotic, (**d**) inhibition of antibiotic by phage, (**e**) trend towards efficacy, but statistical significance not met (no effect), (**f**) no trend seen (no effect). In (**a**,**b**), two different clinical strains demonstrate the additive effects of combining individual antibiotics and phage. (**a**). Strain 7 demonstrated synergy of effect in the addition of phage to tobramycin at in planktonic culture. (**b**). Strain 5 demonstrated the additive effects in the addition of phage with ceftazidime in biofilm culture. In (**c**,**d**), two strains demonstrated different types of inhibition when combining individual antibiotics and phage. (**c**). Strain 3 in planktonic culture demonstrated inhibition of the action of phage when ceftazidime was added. (**d**). Strain 5 in biofilm culture demonstrated antagonism of the action of tobramycin when phage was added. In (**e**,**f**), two examples where there was no significant effect of the antibiotic-phage combination. (**e**). Strain 14 in biofilm culture demonstrated more biofilm eradication with the combination of ceftazidime-phage, but these results did not meet statistical significance. (**f**). Strain 14 in biofilm culture demonstrated no effect on biomass reduction with a combination of tobramycin-phage. Results depicted are for the means of 3 biological replicates combining 6 technical replicates in each group. One-way ANOVAs correcting for multiple comparisons; * *p* < 0.05, ** *p* < 0.01, *** *p* < 0.001.

**Table 1 antibiotics-12-00593-t001:** Summary data for checkerboard experiments across 10 clinical strains + laboratory strain PA01.

Antibiotic	Ceftazidime		Tobramycin	
Culture Type	Planktonic	Biofilm	Planktonic	Biofilm
PA01	A	S	A	N
Strain 1	N	N	N	N
Strain 3	I *	N	N	N
Strain 4	A	N	N	N
Strain 5	A	A	A	I ^#^
Strain 7	A	A	S	N
Strain 9	N	N	N	N
Strain 10	N	A	N	N
Strain 11	A	S	A	N
Strain 14	A	N	A	N
Strain 15	A	A	N	I *

Table 1 summarises the results for both planktonic and biofilm assays, for which there were 3 biological replicates comprising 6 technical replicates in each treatment group. To qualify as anything but N (No added benefit of combination antibiotic + phage), the means of 3 biological replicates had to meet statistical significance using one-way ANOVA correcting for multiple comparisons. A = additional benefit for the combination of antibiotic + phage; S = synergistic benefit to adding antibiotic + phage; I = inhibitory effect of the combination of antibiotic + phage. Definitions of A, N, S and I in Methods. Colour coding: dark green = synergistic; light green = additive; yellow = no significant benefit; red = antagonistic. * indicates antibiotic inhibited the efficacy of phage; # indicates that phage was inhibitory to the effects of the antibiotic.

## Data Availability

More detailed data from this manuscript can be found by accessing the MD(Res) thesis online through the Spiral portal through Imperial College London (https://spiral.imperial.ac.uk/, accessed on 1 December 2022). The thesis is entitled “Lytic bacteriophage as an adjunct to antibiotic therapy for *Pseudomonas aeruginosa* lung infection in cystic fibrosis patients”.

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
