# Peer review of "Lytic Bacteriophage Is a Promising Adjunct to Common Antibiotics across Cystic Fibrosis Clinical Strains and Culture Models of Pseudomonas aeruginosa Infection"

_antibiotics, 2023, doi:10.3390/antibiotics12030593_

Round 1

Reviewer 1 Report

The authors have written a research article titled “Lytic bacteriophage is a promising adjunct to common antibiotics across cystic fibrosis clinical strains and culture models of Pseudomonas aeruginosa infection”. The manuscript must be thoroughly revised, along with some sentence framing, spelling error and grammar check. The manuscript must be corrected with consideration following points.

1.      Line 11., please correct as “antimicrobials”

2.      Line 19, “was added to a 2x the published breakpoint” please make it clear

3.      Please correct “The 4 most antibiotic” as “The four most antibiotic”.

4.      Please rewrite “this success has required chronic daily suppressive therapy nebulised into the lungs”

5.      Please enrich the introduction section with the recent relevant studies

6.      Please do statistical comparison of the results as per levels of the significance

7.      Phages were obtained from which source and how?

8.      Please remove the references from the conclusion section

9.      Inclusion of the zone of the inhibition of the bacterial petri plate or other data may enhance the readability of manuscript.

Author Response

We thank the reviewer for these thoughtful comments. Below, we describe how we have addressed each point.

Reviewer 1:

The authors have written a research article titled “Lytic bacteriophage is a promising adjunct to common antibiotics across cystic fibrosis clinical strains and culture models of Pseudomonas aeruginosa infection”.  The manuscript must be corrected with consideration following points.

  1. Line 11., please correct as “antimicrobials”

Instead of reading “phages are an antimicrobial”, it now reads “phages are antimicrobials”.

  1. Line 19, “was added to a 2x the published breakpoint” please make it clear

This line now clarifies (ceftazidime 16mg/ml + tobramycin 8mg/ml).

  1. Please correct “The 4 most antibiotic” as “The four most antibiotic”.

Corrected.

  1. Please rewrite “this success has required chronic daily suppressive therapy nebulised into the lungs”

Rewritten as: “this requires chronic daily suppressive therapy nebulised into the lungs”.

  1. Please enrich the introduction section with the recent relevant studies

In addition to the studies that are currently cited in the introduction (phage antibiotic antagonism references 19 (2017) in Pseudomonas aeruginosa and rerence 17 (2017) in a different bacterial model, we have now added the Gu liu study (2020) that was included in the discussion discussing synergy + and Tkhilaishvili (2018). Also a review of phage antibiotic interactions - Abedon (2019). Further more recent studies are in the discussion.

  1. Please do statistical comparison of the results as per levels of the significance

These statistical tests had been done and were alluded to in the text, but for some did not appear in the figures and tables.

For Figure 1 (b+d), we have now added the asterisks denoting the level of statistical significance. Example: “ (***, p < 0.001 using the Wilcoxon signed rank test)”

For Table 1, we have added: “To qualify as anything but N (No added benefit of combination antibiotic + phage), means of 3 biological replicates had to meet statistical significance using a one way ANOVA correcting for multiple comparisons (p<0.05).”

  1. Phages were obtained from which source and how?

We have added a section on this in methods.

“The phages used in this study were provided by AmpliPhi Australia Pty Ltd (Brookvale, NSW 2100, Australia). AB-PA01 is a combination of four lytic bacteriophages, Pa193 and Pa204 belonging to the Myoviridae family, and Pa222 and Pa223 belonging to the Podoviridae family. The titer for the phage mix on the company's P. aeruginosa strains used for these experiments was 4x108 PFU/mL. The phage titre was determined using plaque assay, as described below. None of these phage components encoded any known bacterial virulence or antibiotic resistance genes, and all phages were considered to be strictly lytic as per the company's published procedures [32]. The phages were produced following the current good manufacturing practice standard (cGMP) and approved by the US Food and Drug Administration as investigational new drug for an expanded access program (NCT03395743) [33]. An updated release of AB-PA01 with higher titre has been used in preclinical studies and compassionate use cases [34, 35].” 

  1. Please remove the references from the conclusion section.

This has been done.

  1. Inclusion of the zone of the inhibition of the bacterial petri plate or other data may enhance the readability of manuscript.

We appreciate this comment, however, the antibiotic sensitivity testing by disc diffusion assay described in the methods section was performed by the Department of Microbiology at the Royal Brompton. They did not provide us with specific zones of inhibition for the strains used in this study; rather, we were provided with information on sensitivity, resistance or intermediate for the agents used, as appropriate. This is now described in more detail in the methods section.

We sincerely hope that these comments have clarified and addressed this reviewer’s suggestions.

Reviewer 2 Report

In vivo studies can give more support for conclusion.

Author Response

We thank the reviewer for these thoughtful comments. Below, we describe how we have addressed ther points.

Reviewer 2:

Comments and Suggestions for Authors

In vivo studies can give more support for conclusion.

Your point was one shared by several reviewers. Specifically, others had wanted to know more information on the phage preparation used in these experiments and whether it was approved or used for human use. While the in vivo safety and efficacy studies in humans are now described in more detail in the manuscript, we have added a new section on the phage and some more lines in the discussion about these compounds having been administered on a compassionate basis anecdotally to good effect.

“None of these phage components encoded any known bacterial virulence or antibiotic resistance genes, and all phages were considered to be strictly lytic as per the company's published procedures [32]. The phages were produced following the current good manufacturing practice standard (cGMP) and approved by the US Food and Drug Administration as investigational new drug for an expanded access program (NCT03395743) [33]. An updated release of AB-PA01 with higher titre has been used in preclinical studies and compassionate use cases [34, 35].” 

We sincerely hope that these comments have clarified and addressed this reviewer’s suggestions.

Reviewer 3 Report

Peer-review report of the research article (antibiotics-2259411)

The manuscript entitled, “Lytic bacteriophage is a promising adjunct to common antibiotics across cystic fibrosis clinical strains and culture models of  Pseudomonas aeruginosa infection” is a good research article submitted for publication in the journal “antibiotics.”

This manuscript needs a major revision before acceptance. 

The authors have precisely described the background for the present study and have explained the results; however, a few discrepancies are present, which need addressing. 

Abbreviations, such as Pa and CF, are used throughout the manuscript. It is advised to use the full name for the first time in each section i.e. introduction, results, etc, and then use the abbreviation where ever required. 

The authors of this manuscript have mentioned the use of a cocktail of phages repeatedly throughout the manuscript; however, the names, nature, morphology, relative concentration, source, and other important information related to such phages is missing. 

The authors have used strain “PA01” in the experiments. Does this strain have special status, or is this similar to the other strains? 

In figure 3, the results of selected strains are given. Does this imply that the results of all other strains are not significant? If not, why only these results are presented?

Line 52, “… phage sensitivity over time for many MDR strains in patients…” Give any example of such MDR strains.  

Lines 134-136, “The 4 strains that demonstrated the greatest level of biofilm reduction with the addition of the phage cocktail (strains 1, 5, 7 and 14) were the most antibiotic-resistant in biofilm culture.” To which antibiotics do these strains show resistance, and what was the extent of resistance? In lines 293-294, the authors explained that the selected strains were sensitive to ceftazidime and tobramycin. 

Lines 168-169. “There was one instance where combination treatment led to reduced antibiotic efficacy…” The combination treatment also lost efficacy for Strain 3 in planktonic culture… Figure 3c. Why is this different from the results of strain 5, figure 3d?

Lines 255-257. “…in practice, antibiotic administration … … … …as high as 10 minutes for individual plates…” In lines 241-246, the authors explained that the order of administration matters. Why did the authors not try to administer phages before antibiotics? 

The conclusion is lacking in defining the best treatment for eradicating Pa strains. 

Reviewer 4 Report

Review reports should contain the following:

  • A brief summary 

The manuscript is about using phage-antibiotic combinations for Pseudomonas treatment. The authors use both high concentrations and sub-inhibition concentrations of both phages and antibiotics to evaluate the effectiveness.

  • General concept comments
    The introduction is clear. The result section is clear. However, all supplemental documents are missing and cannot be downloaded (line 434-435).
  • Method:

Need to clarify the phage combination. The authors wrote that the phage and cocktails were previously demonstrated. However, it would be nice to know more about the cocktail. What is the ratio of each phage in the cocktail? An author should mention how much MOI has been used in the experiments.

Result:

Please discuss the differences between the percent reduction of biofilm by 2 methods, crystal violet, and resazurin assay. The results from 2 methods were not agreeable in certain strains.

Author Response

We thank the reviewer for these thoughtful comments. Below, we describe how we have addressed each point.

Reviewer 4:

We thank the reviewer for these thoughtful comments. Below, we describe how we have addressed each point.

General concept comments

The introduction is clear. The result section is clear. However, all supplemental documents are missing and cannot be downloaded (line 434-435).

We did not realise that this was the case with the original submission. These elements have now been brought into the main text. In particular, there is now a section in methods that addresses your next point.

Method:

Need to clarify the phage combination. The authors wrote that the phage and cocktails were previously demonstrated. However, it would be nice to know more about the cocktail. What is the ratio of each phage in the cocktail? An author should mention how much MOI has been used in the experiments.

As more than one reviewer has requested this information, it has been moved into the main text under methods.

“The phages used in this study were provided by AmpliPhi Australia Pty Ltd (Brookvale, NSW 2100, Australia). AB-PA01 is a combination of four lytic bacteriophages, Pa193 and Pa204 belonging to the Myoviridae family, and Pa222 and Pa223 belonging to the Podoviridae family. The titer for the phage mix on the company's P. aeruginosa strains used for these experiments was 4x108 PFU/mL. The phage titre was determined using plaque assay, as described below. None of these phage components encoded any known bacterial virulence or antibiotic resistance genes, and all phages were considered to be strictly lytic as per the company's published procedures [32]. The phages were produced following the current good manufacturing practice standard (cGMP) and approved by the US Food and Drug Administration as investigational new drug for an expanded access program (NCT03395743) [33]. An updated release of AB-PA01 with higher titre has been used in preclinical studies and compassionate use cases [34, 35].” 

Result:

Please discuss the differences between the percent reduction of biofilm by 2 methods, crystal violet, and resazurin assay. The results from 2 methods were not agreeable in certain strains.

The rationale in performing two assays was that they both assess different aspects of the biofilm: biomass and viability.

In the methods section, this has been expanded: “To assess biofilm reduction, we used 96-well plates for two different, high-throughput assays: crystal violet (CV) to assess biofilm biomass [40] and resazurin salt (Alamar Blue, IUPAC name: 7-hydroxy-10-oxiphenoxazin-10-ium-3-one) to assess cell viability [41].”

In the results of our experiments, these assays tended to correlate well with each other in general, however, as they measure different things, the correlation was not a perfect one: “Results of the resazurin assay (cell viability) and of the crystal violet (CV) assay (biomass) correlated well with each other (Pearson coefficient 0.80, p=0.0002).”

Also, some further analysis included in the discussion:

“… notably the (CV) assay does not give an approximation of cell viability and [an increase in biomass] could have simply been more dead matter that had adhered to the well. Although the results of the CV and resazurin assays correlated well, this was not true for every strain.” 

We sincerely hope that these comments have clarified and addressed this reviewer’s suggestions.

Reviewer 5 Report

Since the results follows after introduction, it is advisable to define the acronyms here, ex. line 111: CV - the term is introduced several rows earlier (line 98) but it is not specified. 

- line 124 - IQ

- line 164 - CDI (is explained first, at line 383)

- line 319 - (PBS0 solution before 180 L of Mueller Hinton broth (...) Maybe it is 180 microL? - to be verified

- line 340 - instead of "I added (...)" - there were added (...)

Author Response

We thank the reviewer for these thoughtful comments. Below, we describe how we have addressed each point.

Reviewer 5:

Since the results follows after introduction, it is advisable to define the acronyms here, ex. line 111: CV - the term is introduced several rows earlier (line 98) but it is not specified. 

This has now been corrected.

- line 124 – IQ

This has now been corrected.

- line 164 - CDI (is explained first, at line 383)

This has now been corrected.

- line 319 - (PBS0 solution before 180 L of Mueller Hinton broth (...) Maybe it is 180 microL? - to be verified

Yes, the reviewer is correct. This has now been changed.

- line 340 - instead of "I added (...)" - there were added (...)

This has also been changed.

We sincerely hope that these comments have clarified and addressed this reviewer’s suggestions.

Round 2

Reviewer 3 Report

The authors of this manuscript have significantly improved the manuscript.